# Revisiting $(\epsilon, \gamma, \tau)$-similarity learning for domain adaptation

Sofien Dhouib

Univ Lyon, INSA-Lyon, Université Claude Bernard Lyon 1, UJM-Saint Etienne, CNRS,
Inserm, CREATIS UMR 5220, U1206, F-69100, LYON, France
sofiane.dhouib@creatis.insa-lyon.fr

Ievgen Redko*

Univ Lyon, UJM-Saint-Etienne, CNRS, Institut d Optique Graduate School Laboratoire
Hubert Curien UMR 5516, F-42023, Saint-Etienne, France
ievgen.redko@univ-st-etienne.fr

## Abstract

Similarity learning is an active research area in machine learning that tackles the problem of finding a similarity function tailored to an observable data sample in order to achieve efficient classification. This learning scenario has been generally formalized by the means of a $(\epsilon, \gamma, \tau)-$good similarity learning framework in the context of supervised classification and has been shown to have strong theoretical guarantees. In this paper, we propose to extend the theoretical analysis of similarity learning to the domain adaptation setting, a particular situation occurring when the similarity is learned and then deployed on samples following different probability distributions. We give a new definition of an $(\epsilon, \gamma)-$good similarity for domain adaptation and prove several results quantifying the performance of a similarity function on a target domain after it has been trained on a source domain. We particularly show that if the source distribution dominates the target one, then principally new domain adaptation learning bounds can be proved.

## 1 Introduction

Many popular supervised learning algorithms rely on pairwise metrics calculated based on the instances of a given data set in order to learn a classifier. For instance, a famous family of k-nearest neighbors algorithms [1] uses distance matrices in order to define the label of a given test point while support vector machines [2] can be extended to handle the non-linear classification using kernel functions. Despite a widespread use of metrics in machine learning, existing distances often do not capture the intrinsic geometry of data with respect to the labels of the available data points. To tackle this problem, the emerging field of metric learning [3, 4] (also known as similarity learning) aims to provide solutions that allow to learn pairwise metrics explicitly from the data, thus making them tailored for the classification or regression problem at hand.

From the theoretical point of view, similarity learning was extensively analyzed in two seminal papers of [5, 6] based on the $(\epsilon, \gamma, \tau)-$good similarity framework for binary classification.

This framework formalizes an intuitive definition of a good similarity function: given a set of landmarks (or reasonable points) of probability mass at least $\tau$, most of data points (a $1 - \epsilon$ probability mass) should be on average more similar to reasonable points of their own class than to points of the opposite class. Based on the proposed formalization, the authors provided performance guarantees for a resulting linear classifier after mapping data into a new feature space defined via the good similarity function. We refer the interested reader to [7] and [8] for other theoretical studies on $(\epsilon, \gamma, \tau)-$ framework in the supervised, and to [9] and [10]) in the semi-supervised learning cases.

While most of the work based on the $(\epsilon, \gamma, \tau)$ framework has been done in the classical context where training and testing data have the same distribution, in several practical scenarios, one may want to transfer the learned similarity function from one domain, usually called source domain, to another, related yet different domain, called target domain. This framework, known as transfer learning, is a notorious research topic in machine learning nowadays [11, 12, 13, 14] often used in situations where the target domain contains few or no labeled instances in order to reduce the time and effort needed for manual labeling or even collecting new data. As many domain adaptation algorithms proposed in the literature are based on metric learning [15, 16, 17], a question about the theoretical guarantees of the general similarity framework naturally arises.

In this paper, we present a theoretical study of the $(\epsilon, \gamma, \tau)-$ framework in the domain adaptation context where only the marginal distributions across the source and the target domains are assumed to change while the labeling functions remain the same[2]. Contrary to the previous works on the analysis of metric learning algorithms in domain adaptation [18, 19], we aim to consider a more general setting without being attached to a particular algorithm in order to investigate to which extent a similarity that is good for a source domain remains good for the target domain. The obtained results are novel in two different ways. First, they provide a complete theoretical study of similarity learning in domain adaptation, a study that has never appeared in the literature before. Second, they show that under certain assumptions on the richness of the source domain with respect to the target one, the target error can be bounded by terms that all explicitly depend on the source domain error.

The rest of the paper is organized as follows. Section 2 presents the learning setting that we consider with some necessary definitions and notations. Section 3 introduces a generalization of the $(\epsilon, \gamma, \tau)$-goodness definition used to provide a theoretical result relating the source and target goodnesses and presents a brief comparison of the obtained bound with the related work. Apart from the source goodness, the established inequality contains a term reflecting the distance between the distributions of two domains and a worst margin term measuring the worst error obtainable by the similarity function for some instance from the learning sample. We analyze the obtained worst margin term in Section 4 and measure the confidence of its empirical estimation. Section 5 is dedicated to the empirical evaluations of the obtained theoretical results. We conclude our paper in Section 6 and give several possible future perspectives of this work.

## 2    Preliminaries

In order to proceed, we first introduce the basic elements related to the $(\epsilon, \gamma, \tau)-$good similarity framework. In what follows, we denote by $\mathcal{X} \subset \mathbb{R}^d$ and $\mathcal{Y} \subset \{-1, 1\}$ the features and labels spaces, respectively. For any real $t$, $t_+$ denotes its positive part, i.e $\max(t, 0)$. As in [6], we define a similarity function as a pairwise function $K : \mathcal{X} \times \mathcal{X} \to [-1, 1]$. We now recall the definition of the $(\epsilon, \gamma, \tau)$-goodness with hinge loss.

**Definition 1** (Balcan et. al. 2008)**.** *A similarity function $K$ is $(\epsilon, \gamma, \tau)$-good in hinge loss for problem (distribution) $\mathcal{P}$ if there exists a (probabilistic) indicator function $R$ of a set of "reasonable points" such that:*

$$\mathop{\mathbb{E}}_{(x,y)\sim\mathcal{P}} \left[ \left( 1 - \frac{y.g(x)}{\gamma} \right)_+ \right] \leq \epsilon, \tag{1}$$

$$\mathop{\mathbb{P}}_{x'\sim\mathcal{P}} \left[ R(x') \right] \geq \tau, \tag{2}$$

*where $g(x) = \underset{(x',y')\sim P}{\mathbb{E}}[y'K(x,x')|R(x')]$.*

In this definition, $\epsilon$ is an upper bound for the expected hinge loss over all the margins $g(x)$, every margin being the average signed similarity of an instance to reasonable points defined by $R$. To control the loss sensitivity to the margin, a division by $0 < \gamma \leq 1$ is applied.

Following this definition, the authors of [6] prove a theorem that guarantees the existence of a linear separator in a new feature space defined via an $(\epsilon, \gamma, \tau)-$good similarity function, a result that is stated by the following theorem.

**Theorem 1** (Balcan et. al. 2008). *Let $K$ be an $(\epsilon, \gamma, \tau)-$good similarity function in hinge loss for a learning problem $\mathcal{P}$. For any $\epsilon_1 > 0$ and $0 < \delta < \frac{\gamma \epsilon_1}{4}$, let $S = \{x'_1, ..., x'_n\}$ be a (potentially unlabeled) sample of size $n = \frac{2}{\tau} \log\left(\frac{2}{\delta}\right)\left(1 + \frac{16}{(\epsilon_1\gamma)^2}\right)$ of landmarks drawn from $\mathcal{P}$. Consider the mapping:*

$$\phi^S : \mathcal{X} \to \mathbb{R}^n$$
$$x \mapsto (K(x, x'_1), ..., K(x, x'_n)).$$

*Then with a probability at least $1 - \delta$ over the draw of $S$, there exists $\beta \in \mathbb{R}^n$ such that:*

$$\underset{(x,y)\sim\mathcal{P}}{\mathbb{E}}\left[\left(1 - \frac{\langle \beta, \phi^S(x)\rangle}{\gamma}\right)_+\right] \leq \epsilon + \epsilon_1. \tag{3}$$

*In other words, the induced distribution $\phi^S(P)$ in $\mathbb{R}^n$ has a linear separator achieving hinge loss at most $\epsilon + \epsilon_1$ at margin $\gamma$.*

One can see this theorem as a variation of the kernel trick used in the SVM algorithm. Indeed, if $K$ is a kernel function and if $\tau = 1$, the expected loss in Equation (3) becomes the non-regularized loss of an SVM defined via kernel $K$. The authors furthermore derive an algorithm from this theorem that minimizes the empirical version of (3), which boils down to a linear programming problem that can be solved efficiently.

# 3 $(\epsilon, \gamma)-$good similarity learning for domain adaptation

In this section, we introduce the main contributions of our paper. We start by giving a definition of $(\epsilon, \gamma)$-goodness with an arbitrary distribution of landmarks, and then propose a generalization bound that relates the goodness of the same similarity function learned on the source and target domains.

## 3.1 Problem setup

For the considered problem, we assume to have access to samples $S$ and $T$ drawn from source and target probability distributions $\mathcal{S}$ and $\mathcal{T}$, respectively. In the context of domain adaptation, $S \subset (\mathcal{X} \times \mathcal{Y})^m$ is labeled whereas $T$ can be partially or totally unlabeled. In the rest of the paper, we suppose that the labeling is deterministic, meaning that there exists a labeling function $f_{\mathcal{S}}$ (resp. $f_{\mathcal{T}}$) such that for every $(x, y)$ in the source domain (resp. in the target domain), $y = f_{\mathcal{S}}(x)$ (resp. $y = f_{\mathcal{T}}(x)$). Hence, we replace every $(x, y) \sim \mathcal{P}$ by writing simply $x \sim \mathcal{P}$ for all probability distributions considered below. Moreover, since we assume that $\underset{(x,y)\sim\mathcal{S}}{\mathbb{P}}[y|x] = \underset{(x,y)\sim\mathcal{T}}{\mathbb{P}}[y|x]$, then we have $f_{\mathcal{S}} = f_{\mathcal{T}}$.

As hinted in [6, Note 2, Theorem 14], the instances and landmarks can be potentially drawn from different distributions. Hence, we propose a modification of Definition 1 given as follows.

**Definition 2.** *A similarity function $K$ is $(\epsilon, \gamma)$-good in hinge loss for problem $(\mathcal{P}, \mathcal{R})$ (where $\mathcal{P}$ is the data distribution whereas $\mathcal{R}$ is the landmarks distribution) if:*

$$\underset{x\sim\mathcal{P}}{\mathbb{E}}\left[\left(1 - \frac{y.g_{\mathcal{R}}(x)}{\gamma}\right)_+\right] \leq \epsilon,$$

*where $g_{\mathcal{R}}(x) = \underset{x'\sim\mathcal{R}}{\mathbb{E}}[y'K(x,x')]$.*

This is a generalization of Definition 1, and the two coincide when we consider the distribution $\mathcal{R}$ defined by $\underset{x \sim \mathcal{R}}{\mathbb{P}} [x \in A] = \underset{x \sim \mathcal{P}}{\mathbb{P}} [x \in A | R(x) = 1]$ for all measurable sets $A$. As for parameter $\tau$, it can be seen as an upper bound for $\underset{x \sim \mathcal{P}}{\mathbb{P}} [x \in \operatorname{supp} \mathcal{R}]$ since in this case, we have $\operatorname{supp} \mathcal{R} \subset \{R(x) = 1\}$. This definition captures the intuition often used to design domain adaptation algorithms as $R$ can be thought of as a "universal landmarks domain" which is independent of the source or target domains. In the case of sentiment classification, for example, it might correspond to negative or positive vocabulary used to express one's opinion independently of the type of the concerned product.

In the rest of the paper, we use the following notations for any data distribution $\mathcal{P}$ and landmark distribution $\mathcal{Q}$. We denote the goodness of $K$ for problem $(\mathcal{P}, \mathcal{Q})$ by

$$\mathcal{E}_{\mathcal{P}, \mathcal{Q}}(K) := \underset{x \sim \mathcal{P}}{\mathbb{E}} \left[ \left( 1 - \frac{y.g_{\mathcal{Q}}(x)}{\gamma} \right)_+ \right].$$

For simplicity, we further denote by $l_\gamma$ the $\gamma$-scaled hinge loss function defined by:

$$l_\gamma : x \mapsto \left( 1 - \frac{x}{\gamma} \right)_+.$$

We let $\mu$ be a probability distribution that dominates all the other probability distributions used afterwards. In addition, $\mathcal{M}_{\mathcal{P}, \mathcal{Q}}(K)$ stands for the worst margin achieved by an element of $x \in \operatorname{supp} \mathcal{P}$ associated with landmark distribution $\mathcal{Q}$, i.e:

$$\mathcal{M}_{\mathcal{P}, \mathcal{Q}}(K) := \sup_{x \in \operatorname{supp} \mathcal{P}} l_\gamma(y g_{\mathcal{Q}}(x)).$$

Note that since $K$ takes values in $[-1, 1]$ (or even if we only assume that $K$ is bounded), $y g_Q (x)$ is also bounded and consequently $l_\gamma(y g_Q (x))$ is bounded thanks to the continuity of $l_\gamma$. This ensures that $\mathcal{M}_{\mathcal{P}, \mathcal{Q}}(K)$ is finite. Finally, if $B$ is a boolean expression, then $[B] := \mathbb{1}_B$ is an indicator of the set on which $B$ holds (Iverson bracket notation).

## 3.2 Relating the source and target goodnesses

Given a similarity function that is $(\epsilon, \gamma)$-good in hinge loss for problem $(\mathcal{S}, \mathcal{R}_1)$, our goal is to bound its goodness on the target set for problem $(\mathcal{T}, \mathcal{R}_2)$, where $\mathcal{R}_1$ and $\mathcal{R}_2$ are not supposed to be equal.

### 3.2.1 Shared landmarks distribution

In order to prepare for a more general result that relates the goodness of a similarity $K$ for problems $(\mathcal{S}, \mathcal{R}_1)$ and $(\mathcal{T}, \mathcal{R}_2)$, we first provide a preparatory result that considers the same landmark distribution $\mathcal{R} = \mathcal{R}_1 = \mathcal{R}_2$. This result is given by the following lemma[3].

**Lemma 1** (same landmarks). *Let $K$ be an $(\epsilon, \gamma)$-good similarity for problem $(\mathcal{S}, \mathcal{R})$. Then $K$ is $(\epsilon + \epsilon', \gamma)$-good for problem $(\mathcal{T}, \mathcal{R})$, where:*

$$\epsilon' = d_{1+, \gamma}(\mathcal{T}, \mathcal{S}) \mathcal{M}_{\mu, \mathcal{R}}(K)$$

*with $d_{1+, \gamma}(\mathcal{T}, \mathcal{S}) = \underset{x \sim \mu}{\mathbb{E}} \left[ \left( \frac{\mathrm{d}\mathcal{T}}{\mathrm{d}\mu} - \frac{\mathrm{d}\mathcal{S}}{\mathrm{d}\mu} \right)_+ [y g_R (x) < \gamma] \right]$. Moreover, if $\mathcal{T} \ll \mathcal{S}$ then the obtained results holds with*

$$\epsilon' = \sqrt{d_{\chi_+^2, \gamma}(\mathcal{T}, \mathcal{S}) \mathcal{M}_{\mathcal{S}, \mathcal{R}}(K)} \sqrt{\epsilon},$$

*where $d_{\chi_+^2, \gamma}(\mathcal{T}, \mathcal{S}) = \underset{x \sim \mathcal{S}}{\mathbb{E}} \left[ \left( \left( \frac{\mathrm{d}\mathcal{T}}{\mathrm{d}\mathcal{S}} - 1 \right)_+ \right)^2 [y g_R (x) < \gamma] \right]$.*

Several observations can be made based on these results. First, we note that the expectation in both divergence terms is taken only on the support of the hinge loss, i.e for instances having a signed margin smaller than $\gamma$, making these terms problem dependent. This dependence is

quite important as it allows to claim that the presented result can be informative in practice. Second, the obtained bounds both contain the term $\mathcal{M}_{\mu,\mathcal{R}}(K)$ which stands for the worst margin achieved by $K$ on some instance of supp $\mu$. In the case of the SVM, this term is analogous to the largest slack variable associated to an instance drawn from the dominating measure $\mu$. For several choices of $\mu$, this term can be difficult to control, as we can estimate it only by observing data drawn from $\mathcal{S}$. This limitation is tackled by assuming that $\mathcal{S}$ dominates $\mathcal{T}$ thus motivating the bounds with $\chi^2$ distance. These latter clearly show the benefit of assuming $\mathcal{T} \ll \mathcal{S}$: the distance term in the bound is multiplied by $\sqrt{\epsilon}$ meaning that having a similarity function achieving a low error on the source domain can leverage the difference between the domains' distributions. Note that the assumption $\mathcal{T} \ll \mathcal{S}$ is quite common in the domain adaptation literature and has already been used in [20]. As mentioned by the authors, it roughly means that the source domain is richer than the target one, an assumption that is quite reasonable in practice.

### 3.2.2 Different landmarks case

We now turn our attention to a more general case where the landmarks distributions vary across two domains. To this end, we assume that a similarity function $K$ is $(\epsilon, \gamma)$-good for $(\mathcal{S}, \mathcal{R}_1)$. Given these assumptions, our goal now is to provide a learning guaranty for the goodness of $K$ for the $(\mathcal{T}, \mathcal{R}_2)$ learning problem. To proceed, we first rewrite the difference between $\mathcal{E}_{\mathcal{T},\mathcal{R}_2}(K)$ and $\mathcal{E}_{\mathcal{S},\mathcal{R}_1}(K)$ as follows:

$$\mathcal{E}_{\mathcal{T},\mathcal{R}_2}(K) - \mathcal{E}_{\mathcal{S},\mathcal{R}_1}(K) = \mathcal{E}_{\mathcal{T},\mathcal{R}_1}(K) - \mathcal{E}_{\mathcal{S},\mathcal{R}_1}(K) + \mathcal{E}_{\mathcal{T},\mathcal{R}_2}(K) - \mathcal{E}_{\mathcal{T},\mathcal{R}_1}(K).$$

By analyzing the obtained expression, we note that the difference between the first two terms can be bounded using Lemma 1 as $\mathcal{E}_{\mathcal{T},\mathcal{R}_1}(K) - \mathcal{E}_{\mathcal{S},\mathcal{R}_1}(K) = \epsilon + \epsilon' - \epsilon = \epsilon'$ where $\epsilon' = \sqrt{d_{\chi^2_+,\gamma}(\mathcal{T},\mathcal{S})\mathcal{M}_{\mu,\mathcal{R}}(K)}\sqrt{\epsilon}$ when $\mathcal{T} \ll \mathcal{S}$ and $d_{1+,\gamma}(\mathcal{T},\mathcal{S})\mathcal{M}_{\mu,\mathcal{R}_2}(K)$ otherwise. Consequently, we further focus solely on the last two terms and, similar to the previous case, provide a result based on both the $L_1$ and $\chi^2$ distances. We prove the following theorem.

**Theorem 2.** *Let $K$ be an $(\epsilon, \gamma)$-good similarity for problem $(\mathcal{S}, \mathcal{R}_1)$. Then $K$ is $(\epsilon+\epsilon'+\epsilon'', \gamma)$-good for problem $(\mathcal{T}, \mathcal{R}_2)$, with $\epsilon'' = \frac{1}{\gamma}d_1(\mathcal{R}_1, \mathcal{R}_2)$ and $\epsilon' = d_{1+,\gamma}(\mathcal{T},\mathcal{S})\mathcal{M}_{\mu,\mathcal{R}_2}(K)$, where $d_1(\mathcal{R}_1, \mathcal{R}_2) = \mathop{\mathbb{E}}_{x' \sim \mu}\left[\left|\frac{\mathrm{d}\mathcal{R}_1}{\mathrm{d}\mu} - \frac{\mathrm{d}\mathcal{R}_2}{\mathrm{d}\mu}\right|\right]$. Moreover, if $\mathcal{T} \ll \mathcal{S}$, then the obtained result holds with $\epsilon' = \sqrt{d_{\chi^2_+,\gamma}(\mathcal{T},\mathcal{S})\mathcal{M}_{\mu,\mathcal{R}}(K)}\sqrt{\epsilon}$.*

The obtained result suggests that it is better to consider the same landmark distribution $\mathcal{R} = \mathcal{R}_1 = \mathcal{R}_2$ for the two domains, as this assumption minimizes the bound by setting $\epsilon'' = \frac{1}{\gamma}d_1(\mathcal{R}_1, \mathcal{R}_2) = 0$. This conclusion is rather intuitive: in many domain adaptation algorithms the source and target domains are aligned using a shared set of invariant components and landmarks can be seen as invariant points allowing to adapt the similarity measure efficiently across domains. For this reason, we focus on the case of a shared landmark distribution in the rest of the paper.

### 3.3 Comparison with other existing results

We now briefly compare the obtained results with some previous related works. To this end, we note that the vast majority of domain adaptation results [21, 22, 23, 18] have the following form

$$\epsilon^l_{\mathcal{T}}(h, f_{\mathcal{T}}) \leq \epsilon^l_{\mathcal{S}}(h, f_{\mathcal{S}}) + d(\mathcal{S}, \mathcal{T}) + \lambda, \tag{4}$$

where $\epsilon^l_{\mathcal{D}}(h, f_{\mathcal{D}}) := \mathop{\mathbb{E}}_{x \sim \mathcal{D}}[l(h(x), f_{\mathcal{D}}(x))|]$ is the error function defined over probability distribution $\mathcal{D}$ for hypothesis and labeling functions $h, f_{\mathcal{D}} : \mathcal{X} \to \mathcal{Y}$ with loss function $l : \mathcal{Y} \times \mathcal{Y} \to \mathbb{R}_+$; $d(\cdot, \cdot)$ is a divergence measure between two domains and $\lambda$ is the non-estimable term related to difficulty of the adaptation task. From (4), we note that our result with $\chi^2$ distance drastically differs from the traditional domain adaptation bounds as, contrary to them, it suggests that source error directly impacts all the terms in the bound. Indeed, the inequality in (4) prompts us to minimize both the source error $\epsilon^l_{\mathcal{S}}$ and the divergence term $d(\mathcal{S}, \mathcal{T})$ assuming that $\lambda$ is small while our result shows that source error given by the goodness

of the similarity function can partially leverage the divergence between the two domains as it multiplies the latter. To the best of our knowledge, the only two other results that have this multiplicative dependence between the source error and the divergence term are [24] and [25], where the variations of Rényi divergence were considered. Contrary to their contributions, our bound involves a divergence term that is restricted to the $[y.g_\mathcal{R}(x) < \gamma]$ set making it intrinsically linked to the considered hypothesis class. Furthermore, we note that the bounds proposed in [25] involve a non-estimable term that, similar to $\lambda$ in (4) is assumed to be small while the worst margin term presented in our result is subject to the analysis provided in the next section.

## 4 Analysis of the worst margin term

As the worst margin term $\mathcal{M}_{\mu,\mathcal{R}}(K)$ is present in both bounds obtained in the previous section, we proceed to its analysis below. It tells us that if there is at least one instance from the source distribution (or from a distribution dominating it) that has a high loss, then the deviation between the target error and the source error is expected to be large. In what follows, we provide an analysis of this term showing first that it can be bounded in terms of $\gamma$ and then presenting a guarantee allowing its finite sample approximation.

### 4.1 A simple bound for the worst margin

A first simple bound for the worst margin term can be obtained as follows:

$$
\mathcal{M}_{\mu,\mathcal{R}}(K) = \sup_{x \in \text{supp}\,\mu} l_\gamma(yg_\mathcal{R}(x)) = \left(1 - \frac{1}{\gamma} \inf_{x \in \text{supp}\,\mu} y.g_\mathcal{R}(x)\right)_+
$$

$$
= \left(1 - \frac{1}{\gamma} \inf_{x \in \text{supp}\,\mu} \mathbb{E}_{x' \sim \mathcal{R}}[yy'K(x,x')]\right)_+ \leq 1 + \frac{1}{\gamma} \leq \frac{2}{\gamma}.
$$

The last inequality comes from the fact that $K : \mathcal{X} \times \mathcal{X} \to [-1,1]$ and that $0 < \gamma \leq 1$. Based on the obtained expression, we note from Lemma 1 that the target goodness can now be bounded in terms of both values that characterize the similarity function in the source domain. On the other hand, replacing the worst margin term in the bound by constant $\gamma$ prevents us from taking it into account when attempting to design a new adaptation algorithm based on the obtained bounds. In this case, it can be useful to estimate this term empirically from the observed data sample by taking the empirical maximum for the source instances and the empirical mean for the landmarks.

### 4.2 An empirical estimation of the worst margin

We intend to measure our confidence in the empirical estimation of the worst margin term by bounding the deviation between the real worst margin term and its empirical counterpart. To this end, we suppose having access to a labeled data sample $S = \{(x_1, y_1), ..., (x_m, y_m)\} \subset (\mathcal{X} \times \mathcal{Y})^m$ drawn from $\mathcal{S}$, inducing an empirical distribution $\hat{\mathcal{S}}$. Similarly, we define a sample $S_\mathcal{R} = \{(x'_1, y'_1), \ldots, (x'_r, y'_r)\}$ and the corresponding empirical distribution $\hat{\mathcal{R}}$. As the notion of the Rademacher complexity is used to establish our result, we give its definition below.

**Definition 3.** *Let $\mathcal{G}$ be a family of mappings from $\mathcal{X}$ to $\mathbb{R}$ and $\mathcal{P}$ be a probability distribution on $\mathcal{X}$. The Rademacher complexity of $\mathcal{G}$ w.r.t. $\mathcal{P}$ and to a sample size $n$ is defined as $\text{Rad}_n(\mathcal{G}) = \mathbb{E}_{S \sim \mathcal{P}^n}\left[\mathbb{E}_\sigma\left[\sup_{g \in \mathcal{G}} \frac{1}{n} \sum_{i=1}^n \sigma_i g(s_i)\right]\right]$, where $\sigma_i$ are independent uniform random variables in $\{-1, +1\}$ called Rademacher random variables and $S = \{s_1, ..., s_n\}$.*

We can now prove the following result.

**Theorem 3.** *Let $K$ be a similarity function defined on a feature space $\mathcal{X}$. Let $\mathcal{M}_{\mathcal{S},\mathcal{R}}(K)$ denote its worst performance associated to loss function $l_\gamma$ and achieved by an example drawn from $\mathcal{S}$, where $\mathcal{R}$ is the landmarks distribution. Assume that $\mathcal{S}$ dominates $\mathcal{T}$ and that the cumulative distribution function $F_{l_\gamma}$ of the loss function associated with $\mathcal{S}$ and $\hat{\mathcal{R}}$ is $k$ times differentiable at $\mathcal{M}_{\mathcal{S},\hat{\mathcal{R}}}(K)$, and that $k > 0$ is the minimum integer such that $F_{l_\gamma}^{(k)} \neq 0$. Then*

*for all $\alpha > 1, r \geq 1$, there exists $m_0 \geq 1$ such that for all $m \geq m_0$, we have with probability at least $1 - \delta$:*

$$\mathcal{M}_{\mathcal{S},\mathcal{R}}(K) \leq \mathcal{M}_{\hat{\mathcal{S}},\hat{\mathcal{R}}}(K) + \frac{2}{\gamma}\operatorname{Rad}_r\left(\mathfrak{H}_1(K)\right) + \frac{1}{\gamma^2}\sqrt{2\frac{\log\left(\frac{2}{\delta}\right)}{r} + \left(\frac{(-1)^{k+1}\log\left(\frac{2\alpha}{\delta}\right)k!}{F_{l_\gamma}^{(k)}(\mathcal{M}_{\mathcal{S},\hat{\mathcal{R}}}(K))m}\right)^{\frac{1}{k}}},$$

*where $\mathfrak{H}_1(K)$ is the hypothesis class defined by $\mathfrak{H}_1(K) = \{x' \mapsto K(x,x'), x \in \operatorname{supp}\mathcal{S}\}$.*

This theorem shows that under certain conditions, the empirical maximum is guaranteed to converge in probability to the real supremum of the distribution's support. The convergence rate depends heavily on the complexity of the similarity function search space represented by the Rademacher complexity term and on the regularity of the loss distribution function reflected by the $m^{-\frac{1}{k}}$ term. This last term dominates the convergence rate when $k > 2$, and we have in general a convergence rate that is $\mathcal{O}(m^{-\frac{1}{\max\{2,k\}}})$.

Due to the bound's dependence on the regularity of $F_{l_\gamma}$, knowing this cumulative distribution function is necessary for an explicit computation of the bound. Furthermore, in the case when $k$ increases, it implies that we may need more data in order to have a truthful estimation of the function's regularity. Thus, this quantity may become non estimable, which goes in line with several other theoretical contributions [18, 21, 22, 23] where the learning bound includes an a priori non estimable term.

## 5  Experiments

The aim of this section is to empirically illustrate the usefulness of the bounds of Lemma 1 on synthetic data set. In what follows, we restrict the similarity search space to the class of bilinear similarity functions parametrized by a matrix $A \in \mathbb{R}^{d \times d}$, where $d$ is the dimension of the feature space, i.e $K(x,x') = K_A(x,x') = \langle x, Ax' \rangle$. This class has been studied in [7] in the context of $(\epsilon, \gamma, \tau)-$goodness and has been shown to benefit from generalization guarantees established based on the algorithmic stability theory [26].

**Data generation**  We generate the source domain data as a set of 500 two-dimensional points drawn from a mixture of two Gaussian distributions with the same isotropic covariance matrices $\sigma^2 I_2$ and mixing coefficients, where $\sigma$ is the chosen standard deviation[4]. Each distribution represents one of the two classes $1$ and $-1$ centered at $(1,1)$ and $(-1,-1)$, respectively. The target data is generated from the same distribution as the source data by rotating clusters' centers by angles varying from $0°$ to $90°$. Examples of the obtained data samples are given in Figure 1. We note that increasing the angle of rotation leads to an increasing divergence between the two domains.

**Algorithmic implementation**  In accordance with Lemma 1, we consider two cases depending on whether $\mathcal{T} \ll \mathcal{S}$ or not and for both we train a similarity function on the generated data sample and search for a weighting function $w : \mathcal{X} \to \mathbb{R}$ such that the bounds are minimized. Note that from theoretical point of view, the support of both distributions is $\mathbb{R}^2$, but in practice the regions that are far from the cluster centers rarely have data points, so the support can be considered in a limited neighborhood around the centers. For both cases, we estimate the divergence term directly from the analytic expression of density of the generating distribution using by calculating a two-dimensional integral.

As in [7], we consider all of the source sample points as landmarks, and denote by $\hat{\mathcal{S}}_W$ the weighted sample empirical distribution defined on $S_W := \{w_i x_i\}_{i=1}^m$ where $W := (w_1, \ldots, w_m)$. Depending on the assumption considered, we aim to solve the following optimization problem:

$$\min_{\substack{W \in \mathbb{R}^m \\ M \geq 0}} \boldsymbol{J}(W,M), \text{ s.t. } M \geq l_\gamma(y_i.g_{\mathcal{S}_W}(x_i)), \ \forall i \in \{1,...,m\}, \tag{5}$$

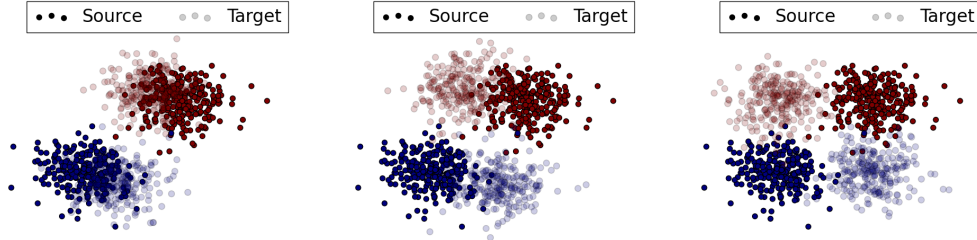

Figure 1: Generated data for (**left**) 30°, (**middle**) 60°, (**right**) 90° degrees rotation.

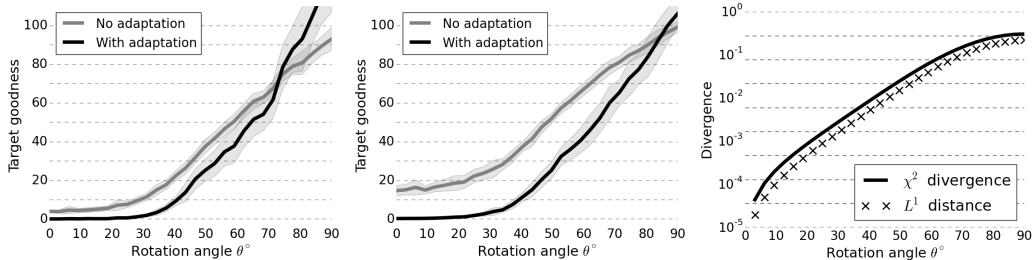

Figure 2: Target goodness as a function of the rotation degree when (**left**) $\mathcal{T} \not\ll \mathcal{S}$ and (**middle**) $\mathcal{T} \ll \mathcal{S}$. For both cases, the similarity function is obtained by solving (5). (**right**) Divergence values for both cases considered. We can observe that rotating the centers of the generating distribution increases both $L^1$ and $\chi^2$ divergences between the samples.

where

$$
\boldsymbol{J}(W, M) = \begin{cases} \mathcal{E}_{\hat{\mathcal{S}},\hat{\mathcal{S}}_W}(K_A) + \sqrt{d_{\chi^2_+,\gamma}(\mathcal{T}, \mathcal{S}) M \mathcal{E}_{\hat{\mathcal{S}},\hat{\mathcal{S}}_W}(K_A)}, & \text{if } \mathcal{T} \ll \mathcal{S}, \\ \mathcal{E}_{\hat{\mathcal{S}},\hat{\mathcal{S}}_W}(K_A) + d_{1+,\gamma}(\mathcal{T}, \mathcal{S}) M, & \text{otherwise.} \end{cases}
$$

**Results**  In Figure 2, we plot the goodness of the similarity function on the target data set before and after adaptation, i.e after solving the minimization problems described above. The results are computed for a rotation angle $\theta$ between 0° and 90°, and after averaging over 30 draws of target samples. From this figure, we can see that the behaviour of the target goodness remains in line with the obtained theoretical results. In both cases considered, optimizing the bounds improves the performance over the "No adaptation" baseline. As expected, in the case of $\mathcal{T} \ll \mathcal{S}$, the target goodness remains lower than when no absolute continuity is assumed due to the minimization of the source error and the worst margin term that impact the entire bound on the target goodness. Note that in the performed empirical evaluations, the divergence term remains constant for every considered rotation angle and is used only as a trade-off parameter. This choice is deliberate as our goal is to show that minimizing the worst margin term and the source error can partially leverage the discrepancy between the two domains. Obviously, the obtained results can be improved by adding a term that properly aligns the two domains distributions through instance-reweighting or feature transformation.

## 6  Conclusions and future perspectives

In this paper, we provided general theoretical guarantees for the similarity learning framework in the domain adaptation context. The obtained results contain a divergence term between the two domains distributions that naturally appears when bounding the deviation between the same similarity's performance on them and a worst margin term measuring the worst error obtainable by the similarity function for some instance from the learning sample. Contrary to the previous generalization bounds established for domain adaptation problem, we showed that when the source distribution dominates the target one, the bound can be improved via a $\sqrt{\epsilon}$ factor. We further analyzed the worst margin term and showed that its

convergence to the true value depends on the complexity of the search space of the similarity function, as well as on the regularity of the hinge loss's cumulative distribution function at a neighborhood of its maximum (worst) value. In order to validate the usefulness of the proposed results, we showed empirically that the minimization of the terms in appearing in the obtained bounds allows to obtain an improved performance over the "no adaptation" baseline without explicitly minimizing the divergence term.

In the future, our work can be extended in multiple directions. First, in our new definition of the $(\epsilon, \gamma)-$goodness, the landmark distribution is assumed to be different from that used to generate source and target data samples and thus a question about the existence of a landmark distribution that leads to tighter bounds naturally arises. Second, it would be interesting to explore the semi-supervised scenario where the landmarks used to learn a similarity function are drawn from the source and target distributions at the same time. In this case, one can expect to obtain a result showing that the goodness of the similarity learned with source landmarks only is worse than that learned on a mixture distribution.

## Acknowledgements

This work benefited from the support provided by the CNRS funding from the Défi Imag'In.

## Footnotes

*The author was at CREATIS when this work was done.

[2]This assumption leads to a setting often called covariate shift problem in domain adaptation.

[3]Due to the limited space, all proofs are provided in the Supplementary material.

[4]In the presented results, we set $\sigma = 0.5$ and provide the same results for other values of $\sigma$ in the Supplementary material.

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
