[Supplementary Material · nips18_metric_camera_ready_supplementary.pdf]

# Supplementary material for the paper: Revisiting $(\epsilon, \gamma, \tau)$-similarity learning for domain adaptation

In this Supplementary material we provide proofs for all theoretical results presented in the main paper and several additional experimental evaluations.

## 1 Proofs from Section 3

Before proving Lemma 1, we first note that one can bound the goodness in the target domain as follows:

$$\mathcal{E}_{\mathcal{T},\mathcal{R}}(K) = \mathcal{E}_{\mathcal{S},\mathcal{R}}(K) + \mathcal{E}_{\mathcal{T},\mathcal{R}}(K) - \mathcal{E}_{\mathcal{S},\mathcal{R}}(K) \leq \epsilon + \mathcal{E}_{\mathcal{T},\mathcal{R}}(K) - \mathcal{E}_{\mathcal{S},\mathcal{R}}(K) \tag{1}$$

following from the $(\epsilon, \gamma)-$goodness of $K$ for $(\mathcal{P}, \mathcal{R})$. Now we focus on the difference between the last two terms in (1). We get the following:

$$\mathcal{E}_{\mathcal{T},\mathcal{R}}(K) - \mathcal{E}_{\mathcal{S},\mathcal{R}}(K) = \mathop{\mathbb{E}}_{x \sim \mathcal{S}} [l_\gamma(y.g_\mathcal{R}(x))] - \mathop{\mathbb{E}}_{x \sim \mathcal{T}} [l_\gamma(y.g_\mathcal{R}(x))] \tag{2}$$

$$= \mathop{\mathbb{E}}_{x \sim \mu} \left[ \frac{\mathrm{d}\mathcal{T}}{\mathrm{d}\mu} l_\gamma(y.g_\mathcal{R}(x)) \right] - \mathop{\mathbb{E}}_{x \sim \mu} \left[ \frac{\mathrm{d}\mathcal{S}}{\mathrm{d}\mu} l_\gamma(y.g_\mathcal{R}(x)) \right]$$

$$\leq \mathop{\mathbb{E}}_{x \sim \mu} \left[ \left( \frac{\mathrm{d}\mathcal{T}}{\mathrm{d}\mu} - \frac{\mathrm{d}\mathcal{S}}{\mathrm{d}\mu} \right)_+ l_\gamma(yg_\mathcal{R}(x))[yg_\mathcal{R}(x) < \gamma] \right], \tag{3}$$

where (3) is obtained by noticing that $t \leq t_+ \; \forall t \in \mathbb{R}$, and due to the positivity of $l_\gamma$ and its property of being zero when calculated at a point $t \geq \gamma$.

We now use (3) as a starting point to prove Lemma 1.

**Lemma 1** (same landmarks)**.** *Let $K$ be an $(\epsilon, \gamma)$-good similarity for problem $(\mathcal{S}, \mathcal{R})$. Then $K$ is $(\epsilon + \epsilon', \gamma)$-good for problem $(\mathcal{T}, \mathcal{R})$, where:*

$$\epsilon' = d_{1+,\gamma}(\mathcal{T}, \mathcal{S}) \mathcal{M}_{\mu,\mathcal{R}}(K)$$

*with*

$$d_{1+,\gamma}(\mathcal{T}, \mathcal{S}) = \mathop{\mathbb{E}}_{x \sim \mu} \left[ \left( \frac{\mathrm{d}\mathcal{T}}{\mathrm{d}\mu} - \frac{\mathrm{d}\mathcal{S}}{\mathrm{d}\mu} \right)_+ [yg_\mathcal{R}(x) < \gamma] \right].$$

*Moreover, if $\mathcal{T} \ll \mathcal{S}$ then the obtained results holds with*

$$\epsilon' = \sqrt{d_{\chi_+^2,\gamma}(\mathcal{T}, \mathcal{S}) \mathcal{M}_{\mathcal{S},\mathcal{R}}(K)} \sqrt{\epsilon}$$

*where*

$$d_{\chi_+^2,\gamma}(\mathcal{T}, \mathcal{S}) = \mathop{\mathbb{E}}_{x \sim \mathcal{S}} \left[ \left( \left( \frac{\mathrm{d}\mathcal{T}}{\mathrm{d}\mathcal{S}} - 1 \right)_+ \right)^2 [yg_\mathcal{R}(x) < \gamma] \right].$$

*Proof.* Using (3), we write:

$$\mathop{\mathbb{E}}_{x \sim \mu} \left[ \left( \frac{\mathrm{d}\mathcal{T}}{\mathrm{d}\mu} - \frac{\mathrm{d}\mathcal{S}}{\mathrm{d}\mu} \right)_+ l_\gamma(y g_\mathcal{R}(x))[y g_\mathcal{R}(x) < \gamma] \right]$$

$$\leq \mathop{\mathbb{E}}_{x \sim \mu} \left[ \left( \frac{\mathrm{d}\mathcal{T}}{\mathrm{d}\mu} - \frac{\mathrm{d}\mathcal{S}}{\mathrm{d}\mu} \right)_+ [y g_\mathcal{R}(x) < \gamma] \right] \mathcal{M}_{\mu,\mathcal{R}}(K) \tag{4}$$

$$= d_{1+,\gamma}(\mathcal{T}, \mathcal{S}) \mathcal{M}_{\mu,\mathcal{R}}(K)$$

where we use Hölder's inequality with $\ell_1$ and $\ell_\infty$ norms to obtain (4). For the case when $\mathcal{S}$ dominates $\mathcal{T}$, we take $\mu = \mathcal{S}$ and we have:

$$\mathop{\mathbb{E}}_{x \sim \mu} \left[ \left( \frac{\mathrm{d}\mathcal{T}}{\mathrm{d}\mu} - \frac{\mathrm{d}\mathcal{S}}{\mathrm{d}\mu} \right)_+ l_\gamma(y g_\mathcal{R}(x))[y g_\mathcal{R}(x) < \gamma] \right]^2 \tag{5}$$

$$= \mathop{\mathbb{E}}_{x \sim \mathcal{S}} \left[ \left( \frac{\mathrm{d}\mathcal{T}}{\mathrm{d}\mathcal{S}} - 1 \right)_+ l_\gamma(y g_\mathcal{R}(x))[y g_\mathcal{R}(x) < \gamma] \right]^2$$

$$\leq \mathop{\mathbb{E}}_{x \sim \mathcal{S}} \left[ \left( \left( \frac{\mathrm{d}\mathcal{T}}{\mathrm{d}\mathcal{S}} - 1 \right)_+ \right)^2 [y g_\mathcal{R}(x) < \gamma] \right] \mathop{\mathbb{E}}_{x \sim \mathcal{S}} \left[ l_\gamma(y g_\mathcal{R}(x))^2 \right] \tag{6}$$

$$= d_{\chi^2_+,\gamma}(\mathcal{T}, \mathcal{S}) \mathop{\mathbb{E}}_{x \sim \mathcal{S}} \left[ l_\gamma(y g_\mathcal{R}(x))^2 \right]$$

$$\leq d_{\chi^2_+,\gamma}(\mathcal{T}, \mathcal{S}) \mathcal{M}_{\mathcal{S},\mathcal{R}}(K) \mathop{\mathbb{E}}_{x \sim \mathcal{S}} [l_\gamma(y g_\mathcal{R}(x))] \tag{7}$$

$$\leq d_{\chi^2_+,\gamma}(\mathcal{T}, \mathcal{S}) \mathcal{M}_{\mathcal{S},\mathcal{R}}(K) \epsilon.$$

To obtain (6), we applied the Cauchy-Schwartz inequality. Inequality 7 is obtained thanks to the boundedness and positivity of $l_\gamma$ via Hölder inequality for norms $\ell_1$ and $\ell_\infty$. The last line follows from the $(\epsilon, \gamma)$−goodness of $K$ for problem $(\mathcal{S}, \mathcal{R})$. □

**Theorem 1.** *Let $K$ be an $(\epsilon, \gamma)$-good similarity for problem $(\mathcal{S}, \mathcal{R}_1)$. Then $K$ is $(\epsilon+\epsilon'+\epsilon'', \gamma)$-good for problem $(\mathcal{T}, \mathcal{R}_2)$, with:*

$$\epsilon'' = \frac{1}{\gamma} d_1(\mathcal{R}_1, \mathcal{R}_2)$$

*and*

$$\epsilon' = d_{1+,\gamma}(\mathcal{T}, \mathcal{S}) \mathcal{M}_{\mu,\mathcal{R}_2}(K),$$

*where $d_1(\mathcal{R}_1, \mathcal{R}_2) = \mathop{\mathbb{E}}_{x' \sim \mu} \left[ \left| \frac{\mathrm{d}\mathcal{R}_1}{\mathrm{d}\mu} - \frac{\mathrm{d}\mathcal{R}_2}{\mathrm{d}\mu} \right| \right]$. Moreover, if $\mathcal{T} \ll \mathcal{S}$, then the obtained result holds with*

$$\epsilon' = \sqrt{d_{\chi^2_+,\gamma}(\mathcal{T}, \mathcal{S}) \mathcal{M}_{\mu,\mathcal{R}}(K)} \sqrt{\epsilon}.$$

*Proof.*

$$\mathcal{E}_{\mathcal{T},\mathcal{R}_2}(K) - \mathcal{E}_{\mathcal{T},\mathcal{R}_1}(K) = \mathop{\mathbb{E}}_{x \sim \mathcal{T}} [l_\gamma(y g_{\mathcal{R}_2}(x)) - l_\gamma(y g_{\mathcal{R}_1}(x))]$$

$$\leq \frac{1}{\gamma} \mathop{\mathbb{E}}_{x \sim \mathcal{T}} [|y g_{\mathcal{R}_1}(x) - y g_{\mathcal{R}_2}(x)|] \tag{8}$$

$$= \frac{1}{\gamma} \mathop{\mathbb{E}}_{x \sim \mathcal{T}} \left[ \left| \mathop{\mathbb{E}}_{x' \sim \mu} \left[ \left( \frac{\mathrm{d}\mathcal{R}_1}{\mathrm{d}\mu} - \frac{\mathrm{d}\mathcal{R}_2}{\mathrm{d}\mu} \right) y y' K(x, x') \right] \right| \right]$$

$$\leq \frac{1}{\gamma} \mathop{\mathbb{E}}_{x \sim \mathcal{T}} \left[ \mathop{\mathbb{E}}_{x' \sim \mu} \left[ \left| \left( \frac{\mathrm{d}\mathcal{R}_1}{\mathrm{d}\mu} - \frac{\mathrm{d}\mathcal{R}_2}{\mathrm{d}\mu} \right) y y' K(x, x') \right| \right] \right] \tag{9}$$

$$\leq \frac{1}{\gamma} \mathop{\mathbb{E}}_{x' \sim \mu} \left[ \left| \frac{\mathrm{d}\mathcal{R}_1}{\mathrm{d}\mu} - \frac{\mathrm{d}\mathcal{R}_2}{\mathrm{d}\mu} \right| \right]. \tag{10}$$

Here (8) holds because $l_\gamma$ is $\frac{1}{\gamma}$−lipschitz. (9) is obtained applying Jensen inequality with the convexity of the $|\cdot|$ function. Line (10) comes from the fact that $|y y' K(x, x')| \leq 1$. As for $\epsilon'$, it is directly obtained by Lemma 1 depending on the assumption made about the absolute continuity of the target distribution with respect to the source distribution. □

## 2 Proof from Section 4

**Theorem 2.** *Let $K$ be a similarity function defined on a feature space $\mathcal{X}$. Let $\mathcal{M}_{\mathcal{S},\mathcal{R}}(K)$ denote its worst performance associated to loss function $l_\gamma$ and achieved by an example drawn from $\mathcal{S}$, where $\mathcal{R}$ is the landmarks distribution. Assume that $\mathcal{S}$ dominates $\mathcal{T}$ and that the cumulative distribution function $F_{l_\gamma}$ of the loss function associated with $\mathcal{S}$ and $\hat{\mathcal{R}}$ is $k$ times differentiable at $\mathcal{M}_{\mathcal{S},\hat{\mathcal{R}}}(K)$, and that $k > 0$ is the minimum integer such that $F_{l_\gamma}^{(k)} \neq 0$. Then for all $\alpha > 1, r \geq 1$, there exists $m_0 \geq 1$ such that for all $m \geq m_0$, we have with probability at least $1 - \delta$:*

$$\mathcal{M}_{\mathcal{S},\mathcal{R}}(K) \leq \mathcal{M}_{\hat{\mathcal{S}},\hat{\mathcal{R}}}(K) + \frac{2}{\gamma}\operatorname{Rad}_r\left(\mathfrak{H}_1(K)\right) + \frac{1}{\gamma^2}\sqrt{2\frac{\log\left(\frac{2}{\delta}\right)}{r}} + \left(\frac{(-1)^{k+1}\log\left(\frac{2\alpha}{\delta}\right)k!}{F_{l_\gamma}^{(k)}(\mathcal{M}_{\mathcal{S},\hat{\mathcal{R}}}(K))m}\right)^{\frac{1}{k}},$$

*where $\mathfrak{H}_1(K)$ is the hypothesis class defined by $\mathfrak{H}_1(K) = \{x' \mapsto K(x,x'), x \in \operatorname{supp}\mathcal{S}\}$.*

*Proof.* To proceed, we first rewrite the quantity of interest as

$$\mathcal{M}_{\mathcal{S},\mathcal{R}}(K) = \mathcal{M}_{\mathcal{S},\mathcal{R}}(K) - \mathcal{M}_{\hat{\mathcal{S}},\hat{\mathcal{R}}}(K) + \mathcal{M}_{\hat{\mathcal{S}},\hat{\mathcal{R}}}(K)$$

and further focus on bounding the difference between the first two terms which can be separated into two quantities as follows:

$$M_1 = \mathcal{M}_{\mathcal{S},\mathcal{R}}(K) - \mathcal{M}_{\mathcal{S},\hat{\mathcal{R}}}(K),$$
$$M_2 = \mathcal{M}_{\mathcal{S},\hat{\mathcal{R}}}(K) - \mathcal{M}_{\hat{\mathcal{S}},\hat{\mathcal{R}}}(K).$$

We begin by bounding $M_1$:

$$M_1 = \sup_{x\in\operatorname{supp}\mathcal{S}} l_\gamma(yg_{\mathcal{R}}(x)) - \sup_{x\in\operatorname{supp}\mathcal{S}} l_\gamma(yg_{\hat{\mathcal{R}}}(x)) \tag{11}$$

$$\leq \sup_{x\in\operatorname{supp}\mathcal{S}}\{l_\gamma(yg_{\mathcal{R}}(x)) - l_\gamma(yg_{\hat{\mathcal{R}}}(x))\} \tag{12}$$

$$\leq \frac{1}{\gamma}\sup_{x\in\operatorname{supp}\mathcal{S}}|g_{\mathcal{R}}(x) - g_{\hat{\mathcal{R}}}(x)| \tag{13}$$

$$= \frac{1}{\gamma}\sup_{x\in\operatorname{supp}\mathcal{S}}\left|\mathbb{E}_{x'\sim\mathcal{R}}[y'K(x,x')] - \frac{1}{r}\sum_{i=1}^r y_i'K(x,x_i')\right|, \tag{14}$$

where (13) holds due to the $\frac{1}{\gamma}$-lipschitzness of $l_\gamma$. The quantity in (14) is known as the representativeness (see, for example, [1]) of sample $S_\mathcal{R}$ drawn from $\mathcal{R}$ associated with the hypothesis set $\mathcal{Y}.\mathfrak{H}_1(K)$. In what follows, we denote it by $\operatorname{Rep}_\mathcal{R}(\mathcal{Y}.\mathfrak{H}_1(K), S_\mathcal{R})$ and notice that its value changes at most by $\frac{2}{\gamma r}$ if an instance of $S_\mathcal{R}$ is replaced since $K$ takes values in $[-1,1]$. By applying Mc-Diarmid's inequality, we have with a probability at least $1 - \frac{\delta}{2}$ for $0 < \delta \leq 1$

$$\operatorname{Rep}_\mathcal{R}(\mathcal{Y}.\mathfrak{H}_1(K), S_\mathcal{R}) \leq \mathbb{E}_{S_\mathcal{R}\sim\mathcal{R}^m}[\operatorname{Rep}_\mathcal{R}(\mathcal{Y}.\mathfrak{H}_1(K), S_\mathcal{R})] + \frac{1}{\gamma}\sqrt{2\frac{\log\left(\frac{2}{\delta}\right)}{r}}. \tag{15}$$

The expectation term in (15) can be bounded by twice the Rademacher complexity of hypotheses class $\mathcal{Y}.\mathfrak{H}_1(K)$ denoted by $\operatorname{Rad}_r\left(\mathcal{Y}.\mathfrak{H}_1(K)\right)$ (see, for example, [1, Lemma 26.2]), which also equals $\operatorname{Rad}_r\left(\mathfrak{H}_1(K)\right)$. Hence, with a probability at least $1 - \frac{\delta}{2}$, we have:

$$M_1 \leq \frac{2}{\gamma}\operatorname{Rad}_r\left(\mathfrak{H}_1(K)\right) + \frac{1}{\gamma^2}\sqrt{2\frac{\log\left(\frac{2}{\delta}\right)}{r}}. \tag{16}$$

Now, we focus on $M_2$ and examine the probability over the draw of $S$ that it exceeds a certain threshold. For a given $t > 0$, we have:

$$\mathop{\mathbb{P}}_{S \sim \mathcal{S}^m} [M_2 \geq t]$$

$$= \mathop{\mathbb{P}}_{S \sim \mathcal{S}^m} \left[ \mathcal{M}_{\mathcal{S}, \hat{\mathcal{R}}}(K) - \mathcal{M}_{\hat{\mathcal{S}}, \hat{\mathcal{R}}}(K) \geq t \right]$$

$$= \mathop{\mathbb{P}}_{S \sim \mathcal{S}^m} \left[ \mathcal{M}_{\hat{\mathcal{S}}, \hat{\mathcal{R}}}(K) \leq \mathcal{M}_{\mathcal{S}, \hat{\mathcal{R}}}(K) - t \right]$$

$$= \mathop{\mathbb{P}}_{S \sim \mathcal{S}^m} \left[ \max_{1 \leq i \leq m} l_\gamma(y_i g_{\hat{\mathcal{R}}}(x_i)) \leq \mathcal{M}_{\mathcal{S}, \hat{\mathcal{R}}}(K) - t \right]$$

$$= \mathop{\mathbb{P}}_{x \sim \mathcal{S}} \left[ l_\gamma(y g_{\hat{\mathcal{R}}}(x)) \leq \mathcal{M}_{\mathcal{S}, \hat{\mathcal{R}}}(K) - t \right]^m$$

$$= F_{l_\gamma} \left( \mathcal{M}_{\mathcal{S}, \hat{\mathcal{R}}}(K) - t \right)^m.$$

By the assumptions made on the regularity of $F_{l_\gamma}$, setting $t$ to $\dfrac{t}{m^{\frac{1}{k}}}$ yields:

$$\mathop{\mathbb{P}}_{S \sim \mathcal{S}^m} \left[ M_2 \geq \frac{t}{m^{\frac{1}{k}}} \right] \tag{17}$$

$$= \left( 1 + F_{l_\gamma}^{(k)}(\mathcal{M}_{\mathcal{S}, \hat{\mathcal{R}}}(K)) \frac{(-t)^k}{m k!} + o\left( \frac{t^k}{m} \right) \right)^m \xrightarrow[m \to \infty]{} \exp\left( F_{l_\gamma}^{(k)}(\mathcal{M}_{\mathcal{S}, \hat{\mathcal{R}}}(K)) \frac{(-t)^k}{k!} \right), \tag{18}$$

where the left-hand side in (18) is obtained from a Taylor expansion. This implies for any $\alpha > 1$ that there exists $m_0 \in \mathbb{N}^*$ such that for all $m \geq m_0$,

$$\mathop{\mathbb{P}}_{S \sim \mathcal{S}^m} \left[ M_2 \geq \frac{t}{m^{\frac{1}{k}}} \right] \leq \alpha \exp\left( F_{l_\gamma}^{(k)}(\mathcal{M}_{\mathcal{S}, \hat{\mathcal{R}}}(K)) \frac{(-t)^k}{k!} \right).$$

Setting this bound to $\frac{\delta}{2}$ and solving for $t$ yields that with a probability at least $1 - \frac{1}{\delta}$

$$M_2 \leq \left( \frac{(-1)^{k+1} \log\left( \frac{2\alpha}{\delta} \right) k!}{F_{l_\gamma}^{(k)}(\mathcal{M}_{\mathcal{S}, \hat{\mathcal{R}}}(K)) m} \right)^{\frac{1}{k}}. \tag{19}$$

Finally we use a union bound to bound the probability that the two inequalities (16) and (19) occur simultaneously in order to obtain the desired result. $\qquad\square$

## 3   Additional experimental evaluations

In this section, we provide experimental results for two data generation scenarios where we vary the variance of the generated classes by setting $\sigma$ equal to 0.2 and 1. We present the generated data in Figure 1. From it, we can observe that increasing the value of $\sigma$ makes the classification problem more difficult. This fact is reflected in Figure 2, where, similar to the main paper, we present the target goodness as a function of the rotation angle for these two scenarios. Several observations can be made from these visualizations. First, the general behaviour of the target goodness remains consistent with that presented in the main paper with the performances of the similarity function obtained by minimizing the established bounds being better that those obtained without adaptation. However, one may also note that the difference between the case $\mathcal{T} \ll \mathcal{S}$ and the case when this assumption is not made becomes less significant. This can be explained by the fact that in these extreme cases, the overall value of the bound is dominated by the source goodness term that remains quite low in the first case and is very high in the second one.

## References

[1] Shai Shalev-Shwartz and Shai Ben-David. *Understanding Machine Learning: From Theory to Algorithms.* Cambridge University Press, New York, NY, USA, 2014.

Figure 1: Generated data for (**left**) 30°, (**middle**) 60°, (**right**) 90° degrees rotation with (**top row**) $\sigma = 0.2$, (**bottom row**) $\sigma = 1$.

Figure 2: Target goodness as a function of the rotation degree when (**left**) $\mathcal{T} \not\ll \mathcal{S}$ and (**middle**) $\mathcal{T} \ll \mathcal{S}$. (**right**) Divergence values for both cases considered. Each row correspond to a different value of $\sigma$ used to generate the data. Here, **top row** $\sigma = 0.2$ and **bottom row** $\sigma = 1$.