[Reviews · NeurIPS 2018]

Reviewer 1



In this paper, the authors establish new bounds for the error in similarity learning for domain adaptation based on information-theoretic notions of distance between the source and target distributions. These establish that a good similarity function chosen for a particular source distribution is also good for a target distribution, as long as (modified versions of) the TV or chi^2 divergence between the two distributions is small. The authors also show that by minimizing the upper bounds they obtain in algorithms for similarity learning, they obtain much better performance over traditional methods. This paper is clearly written and appears to be correct, and the bounds are relatively "user friendly." It does not strike me as a huge breakthrough, but it does contain several nice ideas, and the empirical results are convincing. One aspect of the paper that could be improved is the treatment of the condition supp T \subset supp S. This is *not* the correct condition; throughout the authors mean that T is absolutely continuous with respect to S, which is not the same thing. (If T is a dirac measure at 1/2 and S is the uniform measure on [0, 1], then supp(T) \subset supp(S), but S does not dominate T.) Likewise, assuming that \mu is a dominating probability distribution is not the same as assuming that supp(mu) contains the support of other probability measures. I do not find Theorem 3 compelling. Bounds of this type are useful to the extent that they can be used to obtain data-dependent estimates of relevant quantities, but this bound is dominated by a term which depends on the derivative of the unknown CDF F_\ell_\gamma and the bound only holds for sample sizes sufficiently large. I would argue that this renders Theorem 3 of little practical or even theoretical interest. Finally, this paper assumes deterministic labels throughout. Can this assumption be relaxed?

Reviewer 2



This paper studies the similarity under the domain adaptation (DA) setting. Authors first gave a new definition of similarity function under DA, and then provided theoretical guarantees for the adaptation problem. Authors also presented theoretical analysis for an important margin term, which can be estimated from finite samples, and it can also be used to design new DA algorithms. Pros: - The paper is well written. The preliminary section on the similarity learning under supervised classification is informative, and it motivates for the work of extending similarity learning to DA problems. - Authors gave two versions of relating similarity for source and target domains: one under the setting of shared landmarks distributions, and one under the setting of different landmark distributions. - Under certain assumptions, the error (given by the similarity study) of source domain has a multiplicative effect on the divergence term, which has certain advantage over standard DA bounds where the error of source domain and the divergence term are additive. - Authors designed a new DA algorithm based on the learning bounds, and demonstrated its effects on synthetic dataset. Cons: - The adaptation result in the different landmarks setting seems less useful. Authors didn't discuss much about the term eps'', which is larger when the landmarks distribution is less similar between source and target domains; and unlike eps', this term is not just restricted to the support of hinge loss. - It will be more interesting to see experiment results on real world applications.

Reviewer 3



This paper is a theoretical look at domain adaptation / transfer learning problems through the lenses of similarity learning. The authors have extended an already established similarity learning theoretical framework to cases where the training and testing distributions differ. The authors rigorously prove the following in this paper: - A (\epsilon,\gamma)-good similarity for a problem in a source domain is also is an (\epsilon + \epsilon', \gamma)-good similarity in a target domain, assuming the same landmark distribution on both the source and the target. - They further extend this result in Theorem 2 when the landmark distributions are different in the source and target domains. In this case, the problem in the target domain becomes (\epsilon + \epsilon' + \epsilon'', \gamma) good. Both \epsilon' and \epsilon'' are formally derived in the paper. The former is a product of the worst margin achieved on some instances and a distance between the source and target distributions. The latter is completely defined by the landmark distributions in the source and target. The authors also provide bounds for the worst margin including another theoretical results (theorem 3) providing a tight upper bound for the true worst margin when estimated empirically. I have found this work very interesting, quite rigorous but difficult to read. I felt that the authors were somehow under pressure to compress the content to fit under the 8 page limit. For instance, in Section 3.2.2, I recommend adding a bit more explanation in lines 162 to 165, explaining how the first difference in the equation in line 161 (or right below it; I think that numbering this equation might be a good idea) balls down to applying lemma 1 and imposes \epsilon' in theorem 2. \epsilon'' is derived from the last two terms in this equation on line 161. While things are all clear with the supplement, I think that putting pointers from the submission to the supplement will help the reader significantly. On line 170, I would refer back to \epsilon'' when discussing why it is a good thing to use the same landmark distributions in the source and target domains. In general, adding more intuition behind the precise mathematical construct will certainly help this paper. The experimental section does illustrate the benefit of model adaptation. However, it would be good to have at least a second use case a bit more complex from the 2D Gaussian one that is proposed with different rotation angles as a way to control the similarities between the source and the target. Beside these comments/suggestions, let me reiterate that I have found the work quite interesting and sound.